# A Study on Information Disorders on Social Networks during the Chilean Social Outbreak and COVID-19 Pandemic

**Marcelo Mendoza** [1,2,3,*], **Sebastián Valenzuela** [2,4,5], **Enrique Núñez-Mussa** [6], **Fabián Padilla** [4,7], **Eliana Providel** [8,9], **Sebastián Campos** [2,9], **Renato Bassi** [2,9], **Andrea Riquelme** [10], **Valeria Aldana** [11] and **Claudia López** [3,9]

1 Department of Computer Science, Pontificia Universidad Católica de Chile, Santiago 8331150, Chile
2 Millennium Institute for Foundational Research on Data, Santiago 7820436, Chile; savalenz@uc.cl (S.V.); sebastian.camposm@sansano.usm.cl (S.C.); renato.bassi@sansano.usm.cl (R.B.)
3 National Center of Artificial Intelligence, Santiago 7820605, Chile; clopez@inf.utfsm.cl
4 School of Communications, Pontificia Universidad Católica de Chile, Santiago 8331150, Chile; padilla.arenas@gmail.com
5 Millennium Nucleus on Digital Inequalities and Opportunities, Santiago 8320155, Chile
6 School of Journalism, Michigan State University, East Lansing, MI 48824, USA; nunezmus@msu.edu
7 Fast Check CL, Santiago 7500000, Chile
8 School of Informatics, Universidad de Valparaíso, Valparaíso 2362905, Chile; eliana.providel@uv.cl
9 Department of Informatics, Universidad Técnica Federico Santa María, Valparaíso 2580128, Chile
10 Faculty of Public Administration, Universidad de Talca, Talca 3471109, Chile; andrea.riquelme@gmail.com
11 Faculty of Public Administration, Universidad San Sebastián, Santiago 7520392, Chile; valeria.aldana@gmail.com
* Correspondence: marcelo.mendoza@uc.cl; Tel.: +56-998790614

**Abstract:** Information disorders on social media can have a significant impact on citizens' participation in democratic processes. To better understand the spread of false and inaccurate information online, this research analyzed data from Twitter, Facebook, and Instagram. The data were collected and verified by professional fact-checkers in Chile between October 2019 and October 2021, a period marked by political and health crises. The study found that false information spreads faster and reaches more users than true information on Twitter and Facebook. Instagram, on the other hand, seemed to be less affected by this phenomenon. False information was also more likely to be shared by users with lower reading comprehension skills. True information, on the other hand, tended to be less verbose and generate less interest among audiences. This research provides valuable insights into the characteristics of misinformation and how it spreads online. By recognizing the patterns of how false information diffuses and how users interact with it, we can identify the circumstances in which false and inaccurate messages are prone to becoming widespread. This knowledge can help us to develop strategies to counter the spread of misinformation and protect the integrity of democratic processes.

**Keywords:** disinformation; fact-checking; information spread; online social networks

## 1. Introduction

Misinformation is an old story fueled by new technologies [1]. As a multifaceted problem that includes constructs such as disinformation, conspiracy theories, false rumors, and misleading content, misinformation is considered as an information disorder [2] that harms the information ecosystem and negatively impacts people's well-being. For example, during the COVID-19 pandemic, anti-vaccine groups used online misinformation campaigns to spread the idea that vaccines are unsafe [3]. As a result, medical and scientific groups trying to combat the pandemic also had to deal with an infodemic [4]. It is important, then, to understand the characteristics and effects of misinformation.

Research in the United States has shown that false information spreads quickly within certain groups [5], especially when it confirms users' existing beliefs. Accordingly, re-

searchers have examined how false information can affect important events such as presidential elections [6] and advertising campaigns [7]. The recent COVID-19 pandemic has shown that misinformation can impact regions such as Europe [8], India [9], and China [10], revealing that this phenomenon has a global reach. Different studies show that misinformation can affect how people view vaccination campaigns [11] and can make it harder for governments to control the spread of the pandemic [12].

Numerous organizations are studying how information disorders impact society [13]. Their objective is to develop public policies that can prevent the harmful effects of misinformation [14]. These efforts demonstrate that misinformation is a multifaceted issue that creates an ecosystem of information disorders [15]. The factors contributing to this ecosystem include producing false content, employing bots to disseminate false information widely [16], and increasing incivility on social media platforms, often fueled by troll accounts [17]. The complexity of the problem means that many questions still need to be answered.

This study aimed to identify the characteristics of misinformation on social media platforms in Chile, a Latin American country where social media usage is widespread and that has experienced several social and political crises that have facilitated the spread of misinformation [18–21]. With a population of 19 million, Chile has 13 million Facebook accounts, 9.7 million Instagram accounts, and 2.25 million Twitter accounts [22]. In 2019, a wave of protests against transportation fares led to a fully fledged social uprising against inequality and the political establishment. With the onset of COVID-19 in 2020, the social outbreak led to a process of drafting a new constitution, which failed in 2021 but was resumed in 2022.

More specifically, the study had two objectives:

– To interpret the dynamics of misinformation propagation on social platforms in Chile. We defined the trajectories and traceability of messages with misinformation spread on Twitter, Facebook, and Instagram. This information will recognize the speed and scope of the propagation, identifying if there are non-evident diffusion patterns.
– To explain and analyze the characteristics of the messages that misinform on social platforms in Chile: within the messages that we studied, we conducted a content analysis, determining which linguistic strategies are used to misinform.

Importantly, we compared the diffusion of stories verified as false, inaccurate, and true by professional fact-checkers; that is, we studied the differences and similarities in the dynamics of misinformation across platforms and between types of content. Our study examined the similarities and differences in misinformation across various topics of our recent history. Specifically, we focused on the Chilean social outbreak, the rise of the COVID-19 pandemic, the beginning of the constitutional process, and the 2021 presidential elections. By exploring these topics, we hope to gain insight into the characteristics of disinformation during different events. We paid close attention to the structural propagation of false information. Additionally, we considered specific properties of the content, including the readability of the texts and accessibility to content from various social media platforms.

This article is organized as follows. We discuss related work in Section 2. The materials and methods used to develop this study are presented in Section 3. The content analysis of Twitter is presented in Section 4. Propagation dynamics on Twitter are discussed in Section 5. The analysis of volumes of reactions on Facebook and Instagram is introduced in Section 6. Finally, we discuss the results and findings of this study in Section 7, and we conclude in Section 8, providing concluding remarks and outlining future work.

## 2. Related Work

Information disorders are a growing concern in the age of social media and digital disruption. Frau-Meigs et al. [14] define information disorders as a result of the "social turn" and the emergence of social media, which has created an ecosystem for disinformation and radicalization. Christopoulou [23] studied the role of social media in the information disorder ecosystem, the initiatives that have been developed, and the taxonomies of false

information types. Wardle [24] argues for smarter definitions, practical, timely empirical research on information disorder, and closer relationships between journalism academics, news organizations, and technology companies. Recent studies [25] have discussed how disinformation has reshaped the relationship between journalism and media and information literacy (MIL), and the challenges faced by MIL and digital journalism in the fight against disinformation. Overall, these studies highlight the importance of understanding and addressing information disorders in the digital age, highlighting that information disorders are a complex and multifaceted issue that requires interdisciplinary collaboration and research.

There are several studies focused on the characterization of misinformation [26] and fake news. These studies aim to identify differences between false, misleading content and true content to assist the work of fact-checkers and help in developing automatic methods for misinformation detection [27,28]. Most of these studies focus on the distinction between false and true news [29], disregarding the analysis of imprecise content (i.e., half-truths). Regarding the variety of characterizations carried out to distinguish between false and true information, some studies focus on linguistic features. For example, Silverman [30] determined that fake news shows higher levels of lexical inconsistency between the headline and the body. Horne and Adali [31] studied various news sources, finding that fake news has linguistic similarities with satirical-style texts. In addition, the study shows that fake news headlines are longer and are generally more verbose. On the other hand, the body of this type of news includes repetitive texts and, typically, are easier to read than real news. Pérez et al. [32] analyzed the text of fake news, focusing on gossip websites. The study shows that fake news uses more verbs than true news. Rashkin et al. [33] introduced a case study based on stories qualified by Politifact (https://www.politifact.com/, accessed on 20 March 2023). They found that fake news uses subjective words to dramatize or sensationalize news stories. They also found that these stories use more intensifiers, comparatives, superlatives, action adverbs, manner adverbs, and modal adverbs than true contents.

Analyzing conversations on Twitter, Mitra et al. [34] showed that discussions around false content generate disbelief and skepticism and include optimistic words related to satire, such as joking or grins. Kumar et al. [35] found that conversations triggered by false news are longer but include fewer external sources than conversations initiated by actual content. Bessi and Ferrara [36] found a significant presence of bots in the spread of fabricated claims on Twitter related to the 2016 US presidential election and found that many of these accounts were newly created.

In relation to the motivations that users of social networks would have to share misinformation, some studies argue that a relevant factor is the confirmation bias [37]. Confirmation bias refers to the tendency to seek, interpret, and remember information in a way that confirms pre-existing beliefs [38]. Essentially, individuals tend to look for information that supports their existing views and overlook information that contradicts them. Confirmation bias can result in the persistence of false beliefs and the reinforcement of stereotypes. According to Pennycook et al. [39], the commonly held belief that people share misinformation due to confirmation bias may be misguided. Instead, their findings suggest a need for more careful reasoning and relevant knowledge linked to poor truth discernment. Moreover, there is a significant disparity between what people believe and what they share on social media, primarily attributable to inattention rather than the intentional dissemination of false information. Since many users unintentionally share fake news, effective interventions can encourage individuals to consider verified news, emphasizing the importance of fact-checking initiatives. Unfortunately, recent studies have highlighted evidence that questions the effectiveness of these initiatives. For example, Ecker et al. [40] surveyed information that illustrates resistance from social network users to correcting information, showing that it is challenging to change their minds after users adopt a stance. In a recent study, van der Linden [28] showed that individuals are prone to misinformation through the illusory truth phenomenon. This effect suggests

that statements that are frequently repeated are more likely to be perceived as true in comparison to statements that are novel or infrequently repeated. As false information can spread rapidly, while fact-checking interventions that verify the facts often come too late, these interventions may not be seen by a significant number of people, leading to their failure to counteract the illusory truth effect.

Regarding propagation dynamics, several studies show that stories labeled as false spread deeper than news labeled as true. For example, Friggeri et al. [41] showed that false content is more likely to be reshared on Twitter, reaching more people than actual news. Zeng et al. [42] showed that so-called fake news spreads faster than true content. Zubiaga et al. [43] analyzed the conversations triggered by various false rumors, finding that, at the beginning, most users engage with the content without questioning it by sharing it or commenting on it. The study shows that questioning this type of information occurs at a later stage and that the level of certainty of the comments remains the same before and after debunking. Finally, Vosoughi et al. [5] showed that the spread of stories fact-checked as false is significantly faster and reaches deeper into the network; in addition, in general, their information cascades are wider than for content veried as true, especially during the early stages of content propagation.

## 3. Materials and Methods

We worked on a collection of content tagged by a media outlet in Chile certified by the International Fact-Checking Network (IFCN) (https://www.poynter.org/ifcn/, accessed on 20 March 2023), Fast Check CL (https://www.fastcheck.cl, accessed on 20 March 2023). This Chilean initiative systematically analyzes the content circulating in national media and social networks. In order to have greater content coverage, we also worked with content verified by two other fact-checking initiatives: Decodificador (https://decodificador.cl/, accessed on 20 March 2023), an independent journalistic initiative, and Fact Checking UC (https://factchecking.cl/, accessed on 20 March 2023), a fact-checking outlet housed in the School of Communications at the Pontificia Universidad Católica de Chile. The three initiatives follow cross-checking verification, board discussion, and source-checking methodologies.

We collected content verified by the three fact-checking agencies mentioned above from October 2019 to October 2021. The content was classified into four thematic dimensions:

- Social outbreak: the Chilean social outbreak refers to the Chilean socio-political crisis that started on 19 October 2019, and, after several months of protests and riots, led to a new political constitution process. This event highlighted the role of media in shaping public opinion and influencing social movements. The use of social media by protesters and the circulation of user-generated content challenged traditional news media's monopoly over information dissemination. The event also demonstrated the need for journalists to report on diverse perspectives and provide critical analysis of events unfolding on the ground.

- COVID-19: COVID-19 reached Chile on 3 March 2020, and resulted in 61,050 deaths and more than 3,000,000 of infections by 2022. The COVID-19 pandemic has been a pivotal event for communication sciences and journalism, revealing both the strengths and weaknesses of the media in disseminating information during a global crisis. It has emphasized the importance of reliable and accurate reporting in a time of uncertainty, as well as the role of journalists as gatekeepers of information. The pandemic has also highlighted the role of social media in disseminating misinformation, leading to the need for increased media literacy and critical thinking skills.

- 2021 Elections: In 2021, Chile held political elections for the congress and the presidency, with the latter featuring a highly polarized contest between right-wing candidate José Antonio Kast and left-wing candidate Gabriel Boric. Despite the intense political climate, Boric emerged victorious by a wide margin in the presidential ballotage. This event is very relevant for the study because the extensive coverage of the candidates by traditional media and social networks has played a critical role

in informing the public. The elections have also raised questions about the use of digital media for political campaigning and the impact of micro-targeting on electoral outcomes. Finally, the elections have brought to light issues of political polarization and the role of the media in contributing to or mitigating these divisions.

- Constitution: On 15 November 2019, the National Congress decided to start a constitutional process in response to the social unrest that arose during the social outbreak. This process involved a plebiscite to ratify the decision, which was approved by 79% of voters. Subsequently, on 15 and 16 May 2021, conventional constituents were elected, and the constitutional process began on 4 July of that same year. The constitutional process has been characterized by public participation, with citizens engaging in debates and discussions about the future of the country. Communication has played a crucial role in facilitating this process, with media organizations providing platforms for dialogue and debate, and social networks enabling citizens to exchange ideas. The constitutional process has brought to the forefront issues of social inequality, human rights, and democratic participation, which have been the subject of intense public debate and discussion. Finally, the constitutional process has emphasized the need for innovative and effective communication strategies to engage citizens and foster public participation in the democratic process.

In addition, other topics were grouped in the category Others. In total, we collected 1000 verified stories, with 130 units of content for Social outbreak, 353 on COVID-19, 79 on 2021 Elections, and 132 on Constitution. Table 1 shows these units of content verified by the fact-checking agency.

**Table 1.** Units of content verified by the fact-checking agencies.

| Topic | Fast Check | Decodificador | Fact Checking UC | Total |
|---|---|---|---|---|
| Social outbreak | 102 | 16 | 12 | 130 |
| COVID-19 | 214 | 53 | 86 | 353 |
| 2021 Elections | 67 | 12 | 0 | 79 |
| Constitution | 57 | 36 | 39 | 132 |
| Other | 216 | 52 | 38 | 306 |
| Total | 656 | 169 | 175 | 1000 |

Of these units of content, 341 (34.1%) have their source traceable to Twitter, 131 (13.1%) to Instagram, 73 (7.3%) to Facebook, 52 (5.2%) to television, 30 (3%) to radio, 35 (3.5%) to WhatsApp, 19 (1.9%) to Youtube, 9 (0.9%) to Tiktok, and the rest (31%) to other sources, including mainstream daily press (12%) and blogs (3.2%), among others.

Each fact-checker works with its verification categories, several of which are equivalent. In order to carry out the homogenization of veracity categories, we reviewed two studies. First, we considered the study by Amazeen [44] to measure the level of agreement between fact-checking projects. This study considers in a single category all of the verifications cataloged with some component of falsehood or imprecision. Following Lim [45], we generated a manual equivalence between the veracity categories. In order to establish the equivalence relationships, we used the descriptions of the qualifications provided by the editors of each fact-checking agency. Accordingly, the research team grouped these categories into three main categories: 'True', 'False', and 'Imprecise'. Imprecise content groups hybrid content, which contains half-truths and outdated content (out of temporal context). Unverifiable units of content were removed from the dataset. Table 2 shows this information per fact-checking agency.

Table 3 shows the units of content verified by thematic dimension. It can be seen that the topic with the most fact-checks is 'COVID-19'. We also observe that the two topics with the highest amount of false labels are COVID-19 and Constitution. There is also a significant amount of imprecise content in these two topics. In both COVID-19 and Constitution, false and imprecise content doubles true content, encompassing 67% in COVID-19 and 71%

in Constitution. Each project gave a different emphasis to the coverage of the topics, but COVID-19 was the most fact-checked one of the three of them.

**Table 2.** Veracity categories per fact-checking agency.

| Veracity | Fast Check | Decodificador | Fact Checking UC | Total |
|---|---|---|---|---|
| True | 284 | 38 | 53 | 375 |
| False | 250 | 86 | 37 | 373 |
| Imprecise | 122 | 42 | 85 | 249 |
| Unverifiable | 0 | 3 | 0 | 3 |
| Total | 656 | 169 | 175 | 1000 |

**Table 3.** Veracity categories per topic.

| Topic | True | False | Imprecise | Unverifiable |
|---|---|---|---|---|
| Social outbreak | 56 | 52 | 22 | 0 |
| COVID-19 | 114 | 136 | 101 | 2 |
| 2021 Elections | 30 | 30 | 19 | 0 |
| Constitution | 38 | 58 | 36 | 0 |
| Other | 137 | 97 | 71 | 1 |

In order to characterize the content, we used the articles from the fact-checking outlets to locate the original content in the social platforms in which they were spread. Accordingly, we retrieved full texts and structural information. In the case of Twitter, we retrieved full text and structural information considering retweets, replies, and comments. For Instagram and Facebook, we only have access to aggregated data provided by the CrowdTangle API (https://www.crowdtangle.com, accessed on 10 March 2022), allowing us to report the total number of interactions per verified content. CrowdTangle is an analytical platform owned by Meta that shares data on users' interactions (i.e., shares, views, comments, likes, and so forth) with public pages, groups, and verified profiles on Facebook and Instagram.

For Twitter, the data collection process was conducted as follows:

– Text data: We searched for related messages on Twitter (the data were collected using https://developer.twitter.com/en/products/twitter-api/academic-research, accessed on 10 March 2022) for each verified original post. We collected replies derived from the original content. We included replies to the original post or its retweets and the full list of interactions derived from them (replies of replies). We implemented this query mechanism using recursion. This query retrieval mechanism allowed us to retrieve a huge number of interactions for each post. Then, for each original post, we consolidated a set of messages that compounds the conversational thread of each verified content.

– Structural data: For a given post, we structured its set of related replies using parent–child edges. The original post was represented as the root of the structure. Each message corresponds to a child of the root node. Interactions between users are direct, so each reply connects to the tweet it mentions. For each message, we retrieved the author's identifier and the post's timestamp. The structure that emerges from this process is a propagation tree, with timestamped edges indicating reactions triggered by the original message.

## 4. Content Analysis in Twitter

We started the content analysis by working on the verified content on Twitter, which corresponds to a subset of the total content included in our dataset. Of the 341 units of content whose source is traceable to Twitter, 307 could be accessed from the Twitter Academic API at the time of data collection (January–March 2022). An essential aspect of the dataset is that the number of tweets associated with each content is quite high.

The dataset comprises 397,253 tweets, with 94,469 replies (23.78%) and 302,784 retweets (76.22%). Table 4 shows the units of content verified on Twitter by topic.

**Table 4.** Veracity categories per topic in Twitter.

| Topic | True | False | Imprecise |
|---|---|---|---|
| Social outbreak | 20 | 15 | 7 |
| COVID-19 | 43 | 37 | 14 |
| 2021 Elections | 7 | 19 | 1 |
| Constitution | 20 | 23 | 29 |
| Other | 39 | 19 | 14 |
| Total | 129 | 113 | 65 |

We calculated a set of linguistic metrics for each content verified on Twitter, together with the messages that comment on it (conversational threads). We reported each feature averaging over the conversational threads retrieved from Twitter. First, we computed the threads' length measured in characters and words. We also counted the number of special symbols/words used in each thread (e.g., emoticons, verbs, nouns, proper nouns, named persons, and named locations, among others). In addition, we computed three polarity-based scores usually used in sentiment analysis: valence, arousal, and dominance. Finally, we computed the Shannon entropy of each sequence of words. For features that count the number of symbols/word occurrences, we reported the feature value relative to the average length of the thread, measured in the number of words, allowing values between subjects to be comparable.

We report these metrics per veracity category in Table 5 and per topic dimension in Table 6. The results in Table 5 show that the units of content related to imprecise claims are larger than the rest of the units of content. In addition, true content tends to be less verbose and more precise in terms of the use of proper nouns, ad-positions, and locations.

**Table 5.** Linguistic features for units of content verified in Twitter per veracity category. Each row shows the feature for the threads retrieved for that topic (on average). Word-level features (rows from emoticons to miscellaneous) are measured as the fraction of the length measured in words. Valence, arousal, and dominance are features in $[0, 10]$ and Shannon entropy in $]-\infty, 0[$. Features marked with * and values in red indicate significant differences. Upward and downward arrows indicate whether the value is higher or lower than that of the other categories. We used one-way ANOVA to compare the means of the three groups and determine if there is a significant difference between them. If the resulting $p$-value from the ANOVA analysis is less than the level of significance (0.05), it is concluded that there is a significant difference between the groups.

| Feature | Range | True | False | Imprecise |
|---|---|---|---|---|
| Length in chars * | $\{0, \infty\}$ | 4171 | 4086 | ↑ **4780** |
| Length in words * | $\{0, \infty\}$ | 687 | 674 | ↑ **785** |
| Valence | $[0, 10]$ | 6.02 | 5.86 | 5.94 |
| Arousal | $[0, 10]$ | 5.28 | 5.28 | 5.26 |
| Dominance | $[0, 10]$ | 5.16 | 5.09 | 5.16 |
| Entropy | $]-\infty, 0[$ | −4.28 | −4.26 | −4.23 |
| Emoticons | $[0, 100]$ | 0 | 0 | 0.04 |
| Verbs * | $[0, 100]$ | ↓ **8.2** | 9.0 | 8.8 |
| Determiners | $[0, 100]$ | 14.1 | 14.4 | 14.2 |
| Nouns | $[0, 100]$ | 20.7 | 20.7 | 20.6 |
| Proper nouns * | $[0, 100]$ | ↑ **13.5** | 12.2 | 12.0 |
| Ad-positions * | $[0, 100]$ | ↑ **18.9** | 17.6 | 17.8 |
| Persons | $[0, 100]$ | 1.4 | 1.4 | 1.3 |
| Locations * | $[0, 100]$ | ↑ **1.8** | 1.4 | 1.5 |
| Organizations | $[0, 100]$ | 1.5 | 1.2 | 1.4 |
| Miscellaneous | $[0, 100]$ | 2.4 | 2.6 | 2.4 |

The results in Table 6 show that the units of content associated with the social outbreak are significantly shorter than the rest. This finding correlates with crisis literacy [46], in which it is reported that conversational threads during crises are characterized by being shorter and triggered by urgency.

The table also shows some linguistic differences. The most notorious (indicated in red) shows that the units of content verified during the '2021 Elections' contain, on average, more proper nouns and mention more persons than the rest of the topics. This result makes sense since it is expected that the units of content of political campaigns mention candidates, which would increase the relative presence of these words in the threads.

**Table 6.** Linguistic features for units of content verified in Twitter per topic dimension. Features marked with * and values in red indicate significant differences. Upward and downward arrows indicate whether the value is higher or lower than that of the other categories. We used one-way ANOVA to compare the means of the five groups at a level of significance = 0.05.

| Feature | Range | Social Out. | COVID-19 | 2021 Elect. | Constitution | Other |
|---|---|---|---|---|---|---|
| Length in chars * | $\{0, \infty\}$ | ↓ **2561** | 4179 | 4907 | 4815 | 4774 |
| Length in words * | $\{0, \infty\}$ | ↓ **422** | 690 | 807 | 783 | 788 |
| Valence | $[0, 10]$ | 5.68 | 5.87 | 5.94 | 6.03 | 6.09 |
| Arousal | $[0, 10]$ | 5.33 | 5.25 | 5.26 | 5.25 | 5.29 |
| Dominance | $[0, 10]$ | 5.05 | 5.12 | 5.15 | 5.16 | 5.17 |
| Entropy | $]-\infty, 0[$ | −4.24 | −4.25 | −4.29 | −4.25 | −4.28 |
| Emoticons | $[0, 100]$ | 0 | 0.02 | 0 | 0 | 0 |
| Verbs | $[0, 100]$ | 8.9 | 8.9 | 8.6 | 8.7 | 8.3 |
| Determiners | $[0, 100]$ | 14.2 | 14.3 | 13.2 | 15.0 | 14.1 |
| Nouns | $[0, 100]$ | 21.2 | 20.8 | 19.8 | 20.6 | 20.6 |
| Proper nouns * | $[0, 100]$ | 12.8 | 11.5 | ↑ **15.0** | 12.0 | 13.3 |
| Ad-positions | $[0, 100]$ | 18.6 | 17.7 | 18.4 | 17.6 | 18.6 |
| Persons * | $[0, 100]$ | 1.5 | 1.1 | ↑ **2.3** | 1.4 | 1.4 |
| Locations | $[0, 100]$ | 1.5 | 1.8 | 1.4 | 1.2 | 1.7 |
| Organizations | $[0, 100]$ | 1.5 | 1.1 | 1.5 | 1.7 | 1.5 |
| Miscellaneous | $[0, 100]$ | 2.5 | 2.4 | 2.5 | 2.6 | 2.5 |

We computed a set of readability metrics (https://py-readability-metrics.readthedocs.io/en/latest/, accessed on 10 March 2022). These metrics measure the linguistic complexity of each conversational thread. The linguistic complexity can be measured differently, such as indicating the level of education necessary to handle the vocabulary that makes up the thread (e.g., GFOG, NDCRS, ARI, FKI, DCRS, and CLI) or the reading ease of a text (e.g., LWF and FRE) [47]. The metrics are based on English resources; then, we translated the conversational threads from Spanish to English, using the Google translate API, to apply these metrics. In addition, we computed two native Spanish readability metrics (IFSZ and FHRI) (https://github.com/alejandromunozes/legibilidad, accessed on 10 March 2022). We computed the following readability metrics:

- GFOG: Gunning's fog index outputs a grade level of 0–20. It estimates the education level required to understand the text. The ideal readability score is 7–8. A value ≥12 is too hard for most people to read.
- NDCRS: The new Dale–Chall readability formula outputs values of 0–20 (grade levels). It compares a text against several words considered familiar to fourth-graders. The more unfamiliar words that are used, the higher the reading level. A value ≤4.9 means grade ≤4, whereas a value ≥10 means grades ≥16 (college graduate).
- ARI: The automated readability index outputs values of 5–22 (age related to grade levels). This index relies on a factor of characters per word. A value of 15–16 means a high school student level (10-grade level).
- FKI: The Flesch–Kincaid index (grade levels) outputs values of 0–18. The index assesses the approximate reading grade level of a text. Level 8 means that the reader needs a grade level ≥8 in order to understand (13–14 years old).

- DCRS: The Dale–Chall readability formula outputs values of 0–20 (grade levels). It compares a text against 3000 familiar words for at least 80% of fifth-graders. The more unfamiliar words that are used, the higher the reading level.
- CLI: The Coleman-–Liau index outputs values of 0–20. The index assesses the approximate reading grade level of a text.
- LWRI: The Linsear Write reading ease index scores monosyllabic words and strong verbs. It outputs values of 0–100, where higher indicates easier.
- FRE: The Flesch reading ease index gives a score between 1–100, with 100 being the highest readability score. Scoring between 70–80 is equivalent to school grade level 8. Higher indicates easier.
- IFSZ: The index of Flesch–Szigriszt standardized for Spanish text [48] outputs values of 1–100. The readability of the text is interpreted as follows: very difficult (<40), somewhat difficult (40–55), normal (55–65), easy (65–80), and very easy (>80).
- FHRI: The Fernández–Huerta readability index for Spanish texts [49] outputs values of 1–100. The readability of the text is interpreted as follows: beyond college (<30), college (30–50), 12th grade (50–60), 8th grade (60–70), 6th grade (70–80), 5th grade (80–90), and 4th grade (90–100).

We show the results of this analysis per veracity category in Table 7.

**Table 7.** Readability metrics per veracity in Twitter. Features marked with * and values in red indicate significant differences. Downward arrows indicate whether the value is lower than that of the other categories. We used one-way ANOVA to compare the means of the three groups at a level of significance = 0.05.

| Metric | Range | True | False | Imprecise |
|--------|-------|------|-------|-----------|
| GFOG * | {0, 20} | 16.12 | 16.11 | ↓ **15.05** |
| NDCRS | {0, 20} | 11.32 | 11.12 | 11.42 |
| ARI * | {5, 22} | 17.12 | ↓ **16.12** | ↓ **15.08** |
| FKI * | {0, 18} | 12.57 | 12.08 | ↓ **10.96** |
| DCRS | {0, 20} | 13.08 | 13.08 | 12.87 |
| CLI * | {0, 20} | 18.64 | ↓ **17.58** | ↓ **17.46** |
| LWRI * | {0, 100} | 8.87 | 8.54 | ↓ **7.02** |
| FRE * | {1, 100} | ↓ **36.12** | 39.65 | 40.09 |
| IFSZ * | {1, 100} | ↓ **64.11** | 68.17 | 67.28 |
| FHRI * | {1, 100} | ↓ **66.31** | 70.12 | 70.14 |

The first six metrics in Table 7 correspond to grade levels. These metrics indicate that the complexity of the texts is in the range of basic-to-medium complexity. We found significant differences between content types when calculating GFOG, ARI, FKI, and CLI. The four metrics consistently indicate that false and imprecise content requires fewer grade levels than true content to be understood. This educational gap between true content and false/imprecise content varies depending on the metric used. For example, the difference between true and imprecise according to the ARI metric is around 2 points, which, in the US grade-level system, is equivalent to a difference between an eleventh-grade student and a ninth-grade student. According to the FKI index (Flesch–Kincaid grade level), the difference between what is true and what is imprecise is also around 2 points. On this scale, the gap moves from 10 to 12, distinguishing a text of basic complexity from one of average complexity. The difference in the other metrics is smaller. The last four indices are ease-of-reading metrics. For all of these indices, imprecise and false content are more accessible to read than true ones. Consistently, these metrics show a gap in readability, both when evaluating texts translated from Spanish to English (LWRI and FRE) and when using the metrics defined with lexical resources in Spanish (IFSZ and FHRI).

We also show the readability scores according to the different topics covered in our study. These results are shown in Table 8.

Table 8 shows that the verified units of content for COVID-19 are less complex than the rest of the topics. These differences are significant for the GFOG, ARI, FKI, and CLI indices, in which the readability gap between this topic and the rest is at least one point. Regarding the ease of reading metrics, the four indices show that the topics with more complex texts are related to the 2021 elections and the constitutional process. On the other hand, the more straightforward texts are associated with COVID-19 and the Chilean social outbreak.

**Table 8.** Readability metrics per topic in Twitter. Features marked with * and values in red indicate significant differences. Downward arrows indicate whether the value is lower than that of the other categories. We used one-way ANOVA to compare the means of the five groups at a level of significance = 0.05.

| Feature | Range | Social Out. | COVID-19 | 2021 Elect. | Constitution | Other |
|---|---|---|---|---|---|---|
| GFOG * | $\{0, 20\}$ | 15.43 | ↓ **15.06** | 16.27 | 16.12 | 15.32 |
| NDCRS | $\{0, 20\}$ | 11.12 | 11.18 | 11.64 | 11.54 | 11.35 |
| ARI * | $\{5, 22\}$ | 16.21 | ↓ **15.87** | 17.56 | 17.42 | 16.45 |
| FKI * | $\{0, 18\}$ | 12.86 | ↓ **10.73** | 12.24 | 12.28 | 11.27 |
| DCRS | $\{0, 20\}$ | 12.64 | 12.98 | 12.46 | 13.82 | 13.65 |
| CLI * | $\{0, 20\}$ | 18.02 | ↓ **17.33** | 18.59 | 18.32 | 17.93 |
| LWRI * | $\{0, 100\}$ | 9.11 | 9.74 | 7.23 | ↓ **6.98** | 9.32 |
| FRE * | $\{1, 100\}$ | 41.23 | 40.08 | ↓ **35.22** | 36.22 | 38.23 |
| IFSZ * | $\{1, 100\}$ | 68.22 | 69.24 | 64.22 | ↓ **63.82** | 65.44 |
| FHRI * | $\{1, 100\}$ | 69.23 | 70.02 | 69.67 | ↓ **68.23** | 69.32 |

## 5. Propagation Dynamics in Twitter

In order to study the propagation dynamics of this content on Twitter, we reconstructed the information cascades for each verified content considering replies, representing 20% of the total interactions triggered by content on Twitter. We released the structural data to favor reproducible research (to access the data, please visit: https://github.com/marcelomendoza/disinformation-data, accessed on 10 March 2023). We used the format suggested by Ma et al. [50] to examine cascades in order to create useful data for professionals and researchers in this field. We focused on three characteristics of the cascades: (1) depth, (2) size, and (3) breadth.

A first analysis calculated these characteristics on all of the threads added according to veracity. The averages of these characteristics and their deviations are shown in Tables 9–11.

Table 9 shows that true content cascades are more shallow than the rest. On the other hand, false contents show trees with a high depth. False and imprecise propagation trees are quite similar. Table 10 shows that false contents reach more people than the rest. This finding corroborates prior results [5], showing that false contents reach more people than other kinds of content. Differences in terms of tree size are significant between the groups. Regarding breadth, Table 11 shows that imprecise propagation trees are wider than the rest.

**Table 9.** Depth of the propagation trees in Twitter (replies). KS-tests: false and true: $D = 0.168$, $p \sim 0.08$, false and imprecise: $D = 0.124$, $p \sim 0.70$, true and imprecise: $D = 0.275$, $p \sim 0.01$. Values in red indicate significant differences. Downward arrows indicate whether the value is lower than that of the other categories. We used one-way ANOVA to compare the means of the three groups at a level of significance = 0.05.

| Depth | Mean | Std | Min | Max |
|---|---|---|---|---|
| True | ↓ **6.92** | 8.54 | 1 | 78 |
| False | 9.13 | 11.54 | 1 | 88 |
| Imprecise | 9.92 | 10.10 | 1 | 60 |

**Table 10.** Size of the propagation trees in Twitter (replies). KS-tests: false and true: $D = 0.206$, $p \sim 0.01$, false and imprecise: $D = 0.152$, $p \sim 0.45$, true and imprecise: $D = 0.195$, $p \sim 0.17$. Values in red indicate significant differences. Upward arrows indicate whether the value is higher than that of the other categories. We used one-way ANOVA to compare the means of the three groups at a level of significance = 0.05.

| Size | Mean | Std | Min | Max |
|---|---|---|---|---|
| True | 485 | 1701 | 3 | 14,237 |
| False | 457 | 669 | 2 | 3570 |
| Imprecise | ↑ **679** | 1101 | 4 | 5321 |

**Table 11.** Breadth of the propagation trees in Twitter (replies). KS-tests: false and true: $D = 0.175$, $p \sim 0.06$, false and imprecise: $D = 0.135$, $p \sim 0.61$, true and imprecise: $D = 0.195$, $p \sim 0.17$. Values in red indicate significant differences. Upward arrows indicate whether the value is higher than that of the other categories. We used one-way ANOVA to compare the means of the three groups at a level of significance = 0.05.

| Breadth | Mean | Std | Min | Max |
|---|---|---|---|---|
| True | 332 | 1235 | 1 | 10,813 |
| False | 299 | 469 | 1 | 2833 |
| Imprecise | ↑ **380** | 543 | 1 | 1883 |

For a second analysis, this time of a dynamic nature, we calculated the information cascades' characteristics as a time function. In this way, it was possible to assess whether there were differences in the propagation dynamics. Based on this analysis, we focused on the growth patterns of the cascades, considering both the depth, size, and breadth of the propagation trees. We show these results in Figures 1, 2, and 3, respectively.

Figure 1 shows that, for a fixed depth of the cascades (the figure marks a depth of 20), the time required to reach that depth depends on the type of content. While imprecise content takes an average of 2 h to obtain that depth, true content takes 15 h. This finding indicates that the contents' speed (depth axis) is conditioned on veracity. Furthermore, the figure shows that imprecise contents acquire a greater depth in less time than false or true contents.

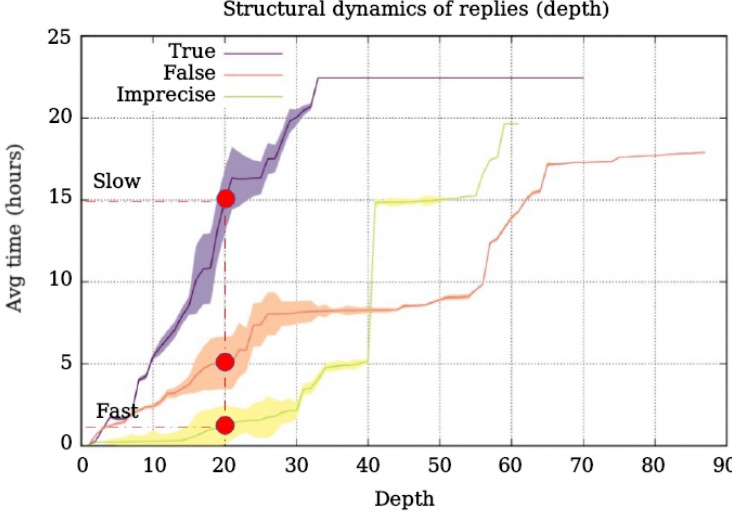

**Figure 1.** Depth growth along time in hours in Twitter.

Figure 2 shows that, for a fixed size (the figure marks what happens for the first 3000 engaged users), the time required to reach that size depends on veracity. While false content takes 2 h to obtain that size, true content needs 6 h. This finding indicates that

the content's growth (users involved in the thread) depends on veracity. Furthermore, the figure shows that the true content infects fewer users than the rest for a fixed observation window. For example, two hours after the original post (see the slice marked as fast), the false content is close to 3000 engaged users on average, whereas the real ones are under 1000. In this dynamic, the imprecise contents are between the true and the false. Finally, the figure shows some real content with many engaged users and a longer permanence than the rest. These viral units of content are few in quantity but significantly impact the network. Our dataset indicates that these units of content correspond to the beginning of the health alert for COVID-19 in Chile, which generated long-lasting threads on the network with many engaged users. In general, this is different from false and imprecise content, which show explosive growth on the network but are more volatile than true content.

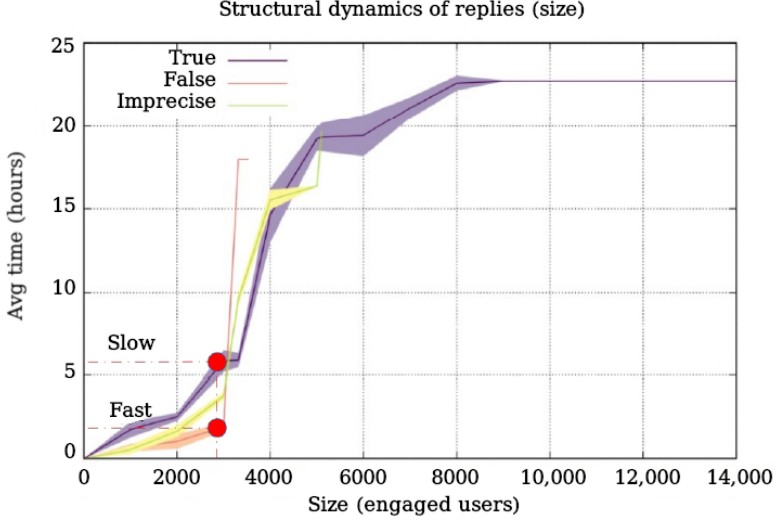

**Figure 2.** Size growth along time in hours in Twitter.

Figure 3 shows that the dynamics of breadth for false and imprecise content are very similar. On the other hand, for true units of content, their cascades grow slower in breadth than the rest. Due to the virality of true content related to COVID-19, the chart of true content grows more than the rest but at a much slower speed than false and imprecise content.

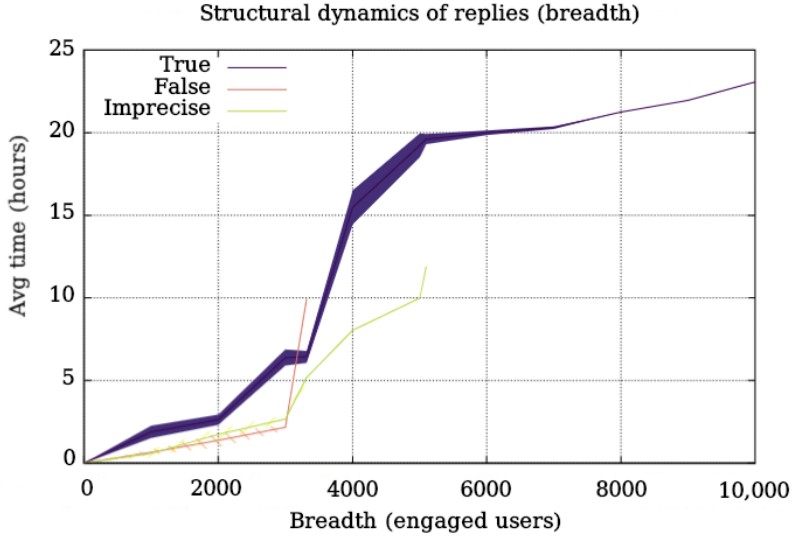

**Figure 3.** Breadth growth along time in hours in Twitter.

## 6. Reactions in Facebook and Instagram

We accessed volumes of reactions from verified content on Facebook and Instagram using the Crowdtangle platform. The platform allows us to know indicators related to volumes but without accessing the text of the messages or structural information. For this reason, the analysis of these two networks is different from the one we were able to carry out on Twitter.

Of the 73 units of content whose source is traceable to Facebook and the 131 traceable to Instagram, 38 and 75 could be accessed from Crowdtangle at the time of data collection (January–March 2022). As Tables 12 and 13 show, Facebook has more false than true content. On the other hand, Instagram has more true than false content.

**Table 12.** Veracity per topic in Facebook (public pages and groups).

| Topic | True | False | Imprecise |
|---|---|---|---|
| Social outbreak | 1 (8.3%) | 1 (5.0%) | 1 (16.6%) |
| COVID-19 | 1 (8.3%) | 7 (35.0%) | 2 (33.3%) |
| 2021 Elections | 5 (41.6%) | 3 (15.0%) | 2 (33.3%) |
| Constitution | 3 (25.0%) | 4 (20.0%) | 0 (0.0%) |
| Other | 2 (16.6%) | 5 (25.0%) | 1 (16.6%) |
| Total | 12 (31.6%) | 20 (52.6%) | 6 (15.8%) |

**Table 13.** Veracity per topic in Instagram.

| Topic | True | False | Imprecise |
|---|---|---|---|
| Social outbreak | 4 (8.8%) | 0 (0.0%) | 2 (10.5%) |
| COVID-19 | 6 (17.6%) | 10 (45.4%) | 4 (21.0%) |
| 2021 Elections | 4 (8.8%) | 2 (9.0%) | 3 (15.7%) |
| Constitution | 6 (17.6%) | 4 (18.0%) | 3 (15.7%) |
| Other | 14 (41.2%) | 6 (27.2%) | 7 (36.8%) |
| Total | 34 (45.3%) | 22 (29.3%) | 19 (25.4%) |

The content formats verified on each network are shown in Tables 14 and 15. Table 14 shows that most of the verified content on Facebook corresponds to photos, while only some content corresponds to external links or videos. Table 15 shows that, on Instagram, most of the content is also photos. However, on Instagram, the volume of verified content corresponding to albums or videos is more relevant than on Facebook.

**Table 14.** Veracity per type of format in Facebook (public pages and groups).

| Type | True | False | Imprecise |
|---|---|---|---|
| Link | 3 (25.0%) | 1 (5.0%) | 1 (16.6%) |
| Status | 1 (8.3%) | 0 (0.0%) | 0 (0.0%) |
| Photo | 8 (66.6%) | 17 (85.0%) | 4 (66.6%) |
| YouTube | 0 (0.0%) | 1 (5.0%) | 0 (0.0%) |
| Live Video | 0 (0.0%) | 1 (5.0%) | 0 (0.0%) |
| Native Video | 0 (0.0%) | 0 (0.0%) | 1 (16.6%) |

**Table 15.** Veracity per type of format in Instagram.

| Type | True | False | Imprecise |
|---|---|---|---|
| Video | 4 (11.7%) | 5 (22.7%) | 3 (15.7%) |
| Photo | 22 (64.7%) | 11 (50.0%) | 15 (78.9%) |
| Album | 8 (23.5%) | 6 (27.2%) | 1 (5.2%) |

The indicators provided by Crowdtangle for verified Facebook pages and groups show volumes of likes, comments, shares, and emotional reactions (e.g., love and wow, among others). We show the total amounts of reactions per veracity type for Facebook in Table 16.

Table 16 shows that fake content attracts more likes than other content. The percentages indicate the volume of likes over the total number of reactions. In false content, likes are equivalent to 37.4% of reactions, whereas, in true content, they reach 20.5% of reactions. In these last units of content, we record more shares than in the false and imprecise contents. Table 16 also shows that the total number of reactions for false content is much higher than the rest. The last row shows the average reaction value per verified content. The average value of reactions for false content is close to 20,000, whereas, for true content, it is only 5401. It is noteworthy that imprecise content produces very few reactions on Facebook.

**Table 16.** Reactions per veracity in Facebook (public pages and groups) Values in red indicate significant differences and upward arrows indicate whether the value is higher than that of the other categories.

| Type | True | False | Imprecise |
|---|---|---|---|
| Likes | 13,315 (20.5%) | ↑ **144,990 (37.4%)** | 1700 (33.6%) |
| Comments | 4935 (7.6%) | 45,543 (11.7%) | 453 (8.9%) |
| Shares | ↑ **35,835 (55.2%)** | 133,555 (34.5%) | 1860 (36.7%) |
| Love | 2369 (3.6%) | 15,084 (3.9%) | 57 (1.1%) |
| Wow | 1165 (1.8%) | 6217 (1.6%) | 249 (4.9%) |
| Haha | 5566 (8.5%) | 29,263 (7.5%) | 593 (11.7%) |
| Sad | 319 (0.5%) | 4739 (1.2%) | 7 (0.1%) |
| Angry | 1153 (1.7%) | 6422 (1.6%) | 124 (2.4%) |
| Care | 163 (0.2%) | 1141 (0.2%) | 15 (0.3%) |
| Total reactions | 64,820 | ↑ **386,954** | 5058 |
| Reactions per content (average) | 5401 | ↑ **19,347** | 843 |

Table 17 shows the reactions to content on Instagram. Crowdtangle provides four reactions: likes, comments, views (only for videos), and shares. Table 17 shows that true content produces more reactions in this network than other types of content. The ratio of these reactions between likes, comments, and shares is almost the same (1/3 for each type of reaction). This finding draws attention since, for false content, many reactions correspond to views, but, for imprecise content, these reactions correspond to likes. Notably, the proportion of comments on Instagram is very low. The last row shows the volume of reactions in proportion to the total verified content. It is observed that true content generates almost twice as many reactions as false and imprecise content (on average).

**Table 17.** Reactions per veracity in Instagram. Values in red indicate significant differences and upward arrows indicate whether the value is higher than that of the other categories.

| Type | True | False | Imprecise |
|---|---|---|---|
| Likes | 1,072,894 (33.9%) | 314,280 (30.2%) | 457,434 42.7%) |
| Comments | 25,942 (0.8%) | 22,608 (2.1%) | 5623 (0.5%) |
| Views (only videos) | 1,026,353 (32.4%) | 416,145 (40.1%) | 151,864 (14.2%) |
| Shares | 1,037,339 (32.8%) | 285,030 (27.4%) | 453,913 (42.4%) |
| Total reactions | ↑ **3,162,528** | 1,038,063 | 1,068,834 |
| Reactions per content (average) | ↑ **93,015** | 47,184 | 56,254 |

## 7. Discussion

This study highlights the characteristics and diffusion process of misinformation in Chile as measured on Twitter, Facebook, and Instagram. The analyses carried out on Twitter considered the content of the messages and propagation dynamics. On Instagram and

Facebook, we had access to volumes of reactions. Several findings emerge from these analyses, which we discuss next:

– On Twitter, true content is less verbose than false or imprecise content (Section 3). We measured that true contents are less verbose than the rest. This finding corroborates the results of Horne and Adali [31], Pérez et al. [32], and Rashkin et al. [33], which show a more significant presence of verbs and their variants in news and comments related to false content. We connect this finding to the study of Mitra et al. [34] , which shows that discussions around false content generate skepticism and include satire, joking, and grins, producing more verbs and verbal expressions than factual content. We also relate this finding to the study of Steinert [17], which connects false content and emotional contagion, a mood that produces verbosity texts. This connection has been explored in several studies illustrating linguistic properties of the encoding of emotions as verbosity [51,52]. However, we found that some conversations triggered by true content are long, with the longest permanence on Twitter, according to our study. This result differs from what was reported by Kumar et al. [35], who showed that the true contents tended to be shorter. One reason for this discrepancy is that the analyzed contents spread during COVID-19, which has its own dynamic not previously observed in other similar studies.

– On Twitter, false and imprecise content show lower reading comprehension barriers than true content (Section 3). We computed various ease-of-reading and grade-level measures, consistently showing lower reading comprehension barriers for false or imprecise content. This finding corroborates the study by Horne and Adali [31], who measured ease-of-reading indices to distinguish between true and false content. In addition, we showed that the access barriers for imprecise content are similar to those for false content. We also connected this finding to the concept of illusory truth [28], which refers to a false statement that is repeated so frequently that it becomes widely accepted as true. This phenomenon is often seen in disinformation campaigns, where false information is deliberately spread to manipulate public opinion. False statements are often written in a simple way so that they are easily remembered and repeated. This can lead to a lowering of reading comprehension standards, as audiences may become more accustomed to simplistic texts and reductionist arguments.

– On Twitter, imprecise content travels faster than true content (Section 4). Our study shows that, in size, as well as in-depth and breadth, true contents are slower than the rest of the content. This finding coincides with the study by Vosoughi et al. [5], who had already shown this pattern in 2018. This study corroborates that this dynamic is maintained on Twitter. This finding can be linked to the connection between emotional contagion and false information [39]. When false information is shared, it often triggers emotional responses and prompts people to take a stance on the issue. This emotional language then trigger interactions between users. Recent studies suggest that these interactions can also lead to a higher level of polarization [53], as the emotional nature of the content may reinforce pre-existing beliefs and opinions.

– On Twitter, false content is reproduced faster than true content (Section 4). Our study also shows that the information cascades of false and inaccurate content grow faster than true content. This finding corroborates the results of Friggeri et al. [41], Zeng et al. [42], and Vosoughi et al. [5], showing that this dynamic is maintained on Twitter. This finding also connects to other studies. For example, King [54] noted that pre-print servers have helped to rapidly publish important information during the COVID-19 pandemic, but there is a risk of spreading false information or fake news. Zhao et al. [55] found that fake news spreads differently from real news even at early stages of propagation on social media platforms such as Weibo and Twitter, showing the importance of bots and coordinated actions in disinformation campaigns.

– On Facebook, false content concentrates more reactions than the rest (Section 5). We connected this finding to two studies. First, Zollo and Quattrociocchi [56] found that Facebook users tend to join polarized communities sharing a common narrative,

acquire information confirming their beliefs even if it contains false claims, and ignore dissenting information. We believe that our observation about volumes of reactions to false content can be triggered from these highly polarized communities. Then, Barfar [57] found that political disinformation in Facebook received significantly less analytical responses from followers, and responses to political disinformation were filled with greater anger and incivility. The study also found similar levels of cognitive thinking in responses to extreme conservative and extreme liberal disinformation. These findings suggest that false contents in Facebook can trigger emotional and ideological responses, producing more interactions between users than factual information.

– On Instagram, true content concentrates more reactions than the rest (Section 5). To the best of our knowledge, this finding is the first result related to Instagram that correlates volumes of reactions and the content's veracity. However, there are studies that offer insights into Instagram users' engagement with content and how brands can utilize the platform to target their audience. Argyris et al. [58] proposed that establishing visual consistency between influencers and followers can create strong connections and increase brand engagement. This suggests that user images on Instagram, as well as text, contribute to engagement, producing higher verification barriers. Ric and Benazic [59] discovered that interactivity on Instagram only influences responses when it is influenced by an individual's motivation to use the application, whether for hedonistic or utilitarian reasons. We believe that since these are the primary motivations on Instagram, informative content receives less attention than on other networks and therefore untrue content is less widely shared.

*Study limitations.* This study has some limitations. Firstly, there was a discrepancy in the availability of data between the three platforms examined. While Twitter provided access to the messages of users who shared false content, Instagram and Facebook only allowed access to statistics on the reactions to the content. Consequently, this study provides a more comprehensive analysis of Twitter and a less detailed analysis of Facebook and Instagram. Due to this disparity, it was not possible for us to track accounts on Instagram and Facebook because Crowdtangle only provides access to aggregated data, while restricting access to individual profiles and other user-specific information. This limitation avoids the analysis of cross-platform spreading and other interesting aspects of the phenomenon. Secondly, the study was hindered by the fact that fact-checking journalism in Chile is still a budding industry, with few verified content and verification agencies available. It is crucial to promote initiatives that combat information disorder, as indicated by the findings of this study. Lastly, this study only investigated the reactions to verified content on social media, without exploring the source of the content or its correlation with traditional media outlets. An investigation that examines the impact of mainstream media narratives on radio and television, and their interplay with social networks, would complement the focus of this study and provide further insights into the causes and effects of information disorder.

## 8. Conclusions

By studying information disorders on social media, this study provides an updated vision of the phenomenon in Chile. Using data collected and verified by professional fact-checkers from October 2019 to October 2021, we find that imprecise content travels faster than true content on Twitter. It is also shown that false content reaches more users than true content and generates more interactions on both Twitter and Facebook. The study shows that Instagram is a network less affected by this phenomenon, producing more favorable reactions to true content than false. In addition, this study shows that access barriers according to ease of reading or grade level are lower for false content than for the rest of the content. It is also shown that true content is less verbose but, at the same time, generates shorter threads of conversation, less depth, and less width than false or imprecise content. These findings suggest that the dynamics of the spread of misinformation vary across platforms and confirm that different language characteristics are associated with false, imprecise, and true content. Thus, future efforts to combat misinformation and

develop methods for the automatic detection of it need to take into account these unique attributes. This is a challenge for media literacy initiatives. In addition, the results invite us to be aware of the context in which the misinformation is produced. By comparing a social outbreak, the COVID-19 pandemic, elections, and the writing of a new constitution, each event shows specific features that frame the false information.

**Author Contributions:** Conceptualization, M.M., S.V., E.N.-M. and F.P.; methodology, M.M.; software, S.C. and R.B.; validation, E.N.-M., F.P., A.R. and V.A.; resources, E.N.-M. and F.P.; data curation, E.N.-M., F.P., E.P., S.C. and R.B.; writing—original draft preparation, M.M.; writing—review and editing, S.V. and E.N.-M.; project administration, C.L. All authors have read and agreed to the published version of the manuscript.

**Funding:** The authors acknowledge funding support from: the National Agency of Research and Development of Chile (ANID), through its Scientific Information Program [grant ANID-PLU200009], the Millennium Institute for Foundational Research on Data (IMFD) [grant ANID-ICN17_002], and the National Center of Artificial Intelligence (CENIA) [grant ANID-FB210017]. Marcelo Mendoza was funded by ANID grant FONDECYT 1200211. Sebastián Valenzuela was funded by the Millennium Nucleus on Digital Inequalities and Opportunities (NUDOS) [grant ANID-NCS2022_046] and ANID grant FONDECYT 1231582. The founders played no role in the design of this study.

**Institutional Review Board Statement:** The study was reviewed by the Research Ethics and Safety Unit of the Pontificia Universidad Católica de Chile and was declared exempt from ethical and safety evaluation, since it will not be investigated with people, personal, and/or sensitive data, nor will living beings or individuals participate in it. Materials, tangible or intangible, specially protected, will be used in scientific research and are not considered risky activities or agents that may put the subjects who participate, carry out, and/or intervene in the research and the environment at risk. The ethics committee certificate is included in the attached documents.

**Informed Consent Statement:** Not applicable.

**Data Availability Statement:** We have released structural data for Twitter content analysis. To access the data, please visit https://github.com/marcelomendoza/disinformation-data (accessed on 20 March 2023).

**Acknowledgments:** We thank the Chilean Transparency Council (CPLT) for promoting this study. We also would like to thank Meta for granting us access to Instagram and Facebook data via the Crowdtangle platform. We also thank Twitter for approving our access to data through the Twitter Academic platform.

**Conflicts of Interest:** The authors declare no conflict of interest. The funders had no role in the design of the study; in the collection, analyses, or interpretation of data; in the writing of the manuscript; or in the decision to publish the results.

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
