# Peer review of "A Study on Information Disorders on Social Networks during the Chilean Social Outbreak and COVID-19 Pandemic"

_applsci, doi:10.3390/app13095347_

Round 1

Reviewer 1 Report

With a very actual topic on information disorder, the paper provides valuable research on the spread of false and inaccurate information online. 

The paper has a good presentation of related work, based on actual references.

The methodology, algorithms, and results are very clear.

Author Response

General comment: With a very actual topic on information disorder, the paper provides valuable research on the spread of false and inaccurate information online. The paper has a good presentation of related work, based on actual references. The methodology, algorithms, and results are very clear.

Reply: Thanks for the positive feedback. This article is the output of a research that took us a lot of time and effort to carry out. We appreciate that you have recognized the effort to carry out this work. Thank you so much.

Comment: English language and style are fine/minor spell check required.

Reply: We carefully reviewed the text of this new version and we believe that we now have an improved text in relation to the original version of the paper. Thanks for your comment.

Reviewer 2 Report

The manuscript studies information disorders on social networks during the Chilean social outbreak and COVID-19 pandemic. The study analyzes data from Twitter, Facebook, and Instagram. The subject of the paper is suitable for publication in the Special Issue of Social Network Analysis: Opportunities and Challenges of the Applied Sciences journal.

The structure of the paper is logical and predictable, which contributes to its readability. The introduction gradually initiates the reader to the topic of the study and at the same time provides the outline of the research. The manuscript does not include any research questions but instead addresses two specific objectives that are treated as research questions.

The literature review is included in section 2 and it is quite short. The next section (Materials and Methods) describes the methodology that is incorporated in the manuscript. The following 3 sections include the results of the study. Sections 4, and 5 refer to results from Twitter, and section 6 reports the results of the study from Facebook and Instagram. Section 7 includes a discussion of the findings which is quite brief if we take into account the number of results that were reported by the study. The manuscript is concluded with a conclusion section that also addresses the issue of future extensions of the study.

The study has a nice organization, adequate methodology, and interesting results. Nevertheless, some issues need to be addressed by the author(s).

The title of the study includes the term information disorder. The term is also mentioned in the conclusion section. I would assume that some literature data on this term would have been included in the introductory section, but this is not the case.  I suggest authors should include a brief discussion of the term along with specific references in section 2.

The study has gathered social data from a specific period. The authors should elaborate on why this particular period was chosen. Some information is provided in the summary of the manuscript but concrete justification needs to be provided in the materials and methods section.

The social content was classified into four thematic dimensions, namely social outbreak, COVID-19, 2021 Elections, New Constitution, and others. Since not many of the journal’s international audience is aware of the social situation in Chile, some reasoning for the selection of these categories should be provided.

The manuscript presents very interesting results. But the discussion section can be characterized as quite brief. There is a connection between the finding with previous studies but the authors do not go into depth with their analysis. I would like to see the discussion section expanded.

Close related to my previous remarks is the fact that there is no mention of possible limitations of the study. I would like to see such a discussion and also how the Chilean context may have an effect on the study findings.

Overall I believe that the study is interesting and if the author(s) address the previously mentioned issues it will be suitable for publication in the Applied Sciences journal.

Author Response

Please check the response letter where we show how each comment was addressed in this revised version of the paper.

Reviewer 3 Report

The authors investigate a very interesting area, of the spread of misinformation in three social networks. Towards this end, a manually annotated multi-topic dataset by three fact-checking agencies is analyzed, and a detailed analysis is presented in one of these topics.

The document is easy to follow, and all related areas are adequately covered. The results are very interesting, and the authors relied on linguistic features to unveil important patterns.

The reviewer feels that this work should be considered for publication, once these recommendations are addressed:

1) Since the numbering format is used for the references, please be concise on the numbering. For example, ref. 13 (line 28) appears after ref. 6 (line 25), whereas ref. 14 is not found in the text.

2) Line 111: It would be great for the readers if a few examples of the “other sources” were named, considering that combined they represent nearly a third of the sources.

3) Line 128: Did the authors mean that the false and imprecise content is double than the true content? Please reconsider the use of “duplicates”.

4) Tables 5 and 6: Please use the full form of the feature, rather than their abbreviated form (e.g., Dets, Propns).

5) Regarding all tables where the outlier values are highlighted, is there a specific threshold for identifying the outlier values (e.g., absolute value, variation)? Can the authors briefly analyze their rationale?

6) Table 16: According to the authors’ highlighting rationale, why the value of the “Reactions per content (average)” of the “False” category is not highlighted as well? Its value is Order of Magnitude larger than the other two.

7) Table 17: As the previous comment, the value of “Total reactions” for the “true” category why it not highlighted?

8) It is a common knowledge, and also reflected by many studies, that a user may maintain accounts in multiple Online Social Networks (OSNs). Since multiple OSNs are evaluated, did the authors investigate whether there are accounts in these OSNs which are essentially maintained by the same entity, spreading false or imprecise news in multiple OSNs? Please comment about that and its implications.

9) For the benefits of the study, the OSN data has been manually annotated by three fact-checking agencies. However, considering that this option is not feasible in the majority of the cases. Can the authors suggest how their work can contribute towards the semi-automatic categorization of the OSN content in the three mentioned categories (True, False, Imprecise)?

Author Response

(The authors gave the same response as above.)

Round 2

Reviewer 2 Report

The authors have addressed all my comments from the first round of review.

Reviewer 3 Report

All the suggested recommendations have been successfully addressed by the authors.

The manuscript should be considered for publication in its current form.